# Social Image Impacting Attitudes of Middle-Aged and Elderly People toward the Usage of Walking Aids: An Empirical Investigation in Taiwan

**DOI:** 10.3390/healthcare8040543

**Published:** 2020-12-09

**Authors:** Shao-Wei Huang, Tsen-Yao Chang

**Affiliations:** 1Graduate School of Design, National Yunlin University of Science and Technology, Yunlin 640, Taiwan; d10530017@gemail.yuntech.edu.tw; 2Department of Creative Design, National Yunlin University of Science and Technology, Yunlin 640, Taiwan

**Keywords:** walking aids, social image, Theory of Reasoned Action (TRA)

## Abstract

The elderly need the assistance of walking aids due to deterioration of their physical functions. However, they are often less willing to use these aids because of their worries about how others may think of them. Not using professional walking aids often makes elderly people fall easily when walking. This study explores the behavioral intention factors of middle-aged people (45–64 years old) and elderly people (65 years and older) that affect the use of walking aids. Based on the Theory of Reasoned Action (TRA), subjective norms, attitude toward usage, behavior intention, safety, and usefulness were combined with social image to establish the research framework. This study used questionnaire surveys both in paper form assisted by volunteers and in online electronic form. A total of 457 questionnaires were collected. Data analysis was carried out in three stages: descriptive analysis, measurement model verification, and structural equation model analysis. The results showed that social image had a significant impact on the attitude toward using walking aids. Factors such as attitude toward usage, subjective norms, and safety of walking aids also had a significant positive impact on behavioral intention. Finally, through the research results, some suggestions are proposed for stakeholders to improve the elderly’s concerns about the social image of using walking aids.

## 1. Introduction

The world is currently facing a serious aging trend, and the speed of aging is rapidly accelerating. For example, it took 150 years for the population over 60 years old in France to go from 10% to 20%. However, it took only around 20 years for countries such as Brazil, China, and India to reach the same level [1]. The “2019 Revision of World Population Prospects” launched by the United Nations pointed out that the world is facing three major trends: a decrease in the new population, a rapid increase in the elderly population, and a decline in population in many countries [2]. According to data from the World Health Organization, in the period from 2015 to 2050, the proportion of the world’s population over 60 will be almost double, from 12% to 22%. The aging population structure has become a major challenge for the entire world. In 2016, the World Health Organization and its members formulated the Global Aging and Health Strategy and Action Plan in the World Health Assembly (Document A69/17) [3]. In addition to focusing on healthy aging, providing a good supportive environment so that the elderly can walk easily is also one of the key points of continuous promotion, which include providing a barrier-free environment or the use of assistive devices. At the same time, the elderly can perform regular exercises to improve their physical and mental functions. It can help the elderly to delay aging [1].

According to the report, “Taiwan Population Estimates (2018–2065)” [4], released by the National Development Council, Taiwan is currently facing three major trends. First, the total population will turn to negative growth in 3–10 years. Second, if the total fertility rate remains the same, the number of births in 2065 will be halved. Third, Taiwan will become an overaged society in the next eight years. The rate of aging has surpassed developed countries such as European nations, America, and Japan. Moreover, once the number of births in the future decreases and the average age of the population increases, this means that the dependency ratio will continue to increase. In addition to major changes in the demographic structure, the social burden will also increase. Moreover, the report pointed out that in the future, the proportion of people over 65 in the total population will rise from 14.5% in 2018 to 41.2% in 2065. This means that four out of 10 people will be elderly [5]. Therefore, in order to deal with the aging social structure, Taiwan is currently actively improving the friendly environment for the elderly, including elderly-friendly workplaces, long-term care programs, and the economic security of the elderly.

The National Health Interview Survey, conducted by Taiwan’s National Health Administration [6], interviewed 3280 elderly people in 2017. It stated that one out of six elderly people fell in the past year (495 people, 15.5%). As the physical functions of the elderly, including muscle strength and balance ability, etc., decline, they are gradually going downhill. Moreover, the survey also mentioned that the most common places to fall at home are the bedroom, living room, and bathroom. Among the top 10 causes of death among the elderly in Taiwan in 2019, accidents ranked sixth [7], even higher than kidney disease or high blood pressure. In addition to deaths in traffic accidents, the most common accidents are caused by falls.

Because of the deterioration of the body, many abilities of the elderly are not as good as when they were young, including vision, hearing, balance, reaction time, muscle strength, or endurance. Yang et al. conducted a study on elderly people with mild to moderate difficulties living in institutions [8]. The survey found that the functions affected by the activities of daily living (ADLs) of these elderly people included climbing stairs (62.9%), bathing (47.2%), and walking (40.4%). At the same time, it was also found that functional movements such as knee extension muscle strength, 3-m timed up and go, 30-s sitting and standing, and 2-min stance stepping were worse than in other normal elderly people.

Therefore, with age the performance of the elderly in ADLs is most often affected by the ability to walk. Suwannarat et al. conducted interviews and surveys on 343 elderly people over 65 years old in rural Thailand in 2015 [9]. The types of walking aids used by these elderly people and their mobile functions were discussed. The elders who participated in the survey had a significant correlation with the need for walking aids, especially the elderly over 75. The ability of daily living can be obviously improved through walking aids.

Some studies have mentioned that the elderly encounter several major obstacles when using assistive technologies, including privacy violations, insufficient trust in technical assistance tools, lack of value-added features, consideration of purchase costs, and lack of training or embarrassment [10]. Related researches tend to focus on high-tech assistive technologies, and rarely mention low-tech walking aids such as canes or walkers. On the topic of preventing the elderly from falling, researchers found out that many elderly people do not want to be associated with health problems. They think that the problem is “not for me” or they are afraid to feel stigmatized after using assistive devices [11]. The elderly do not want to use assistive devices because of their worry about dependence [12].

Simpson and Richardson [13] suggested that when a patient needs to use walking aids, medical staff should actively check the walking aids used by the patient and his or her health promotion needs. At the same time, the elderly are encouraged to seek professional advice before using walking aids. However, it can be observed in Taiwan that there are still many elders who have difficulty in moving and are reluctant to use walking aids such as canes or walkers, whether or not they need disability care services or are provided assistive device purchase subsidies. They are concerned about their social image and are afraid of being laughed at by others or being labeled as “disabled.” They would rather struggle to walk or use umbrellas to assist themselves in walking. There is still a lack of relevant empirical research on the phenomenon of elderly people resisting the use of walking aids from the perspective of social image.

In 2014, the Technology Acceptance Model (TAM) was used to study the technology acceptance of the elderly in Hong Kong [14]. The results of the study pointed out that because Chinese people are more introverted and afraid of being laughed at for poor performance, it is easy to influence the intention of the elderly to use technology. In addition, Tural, Lu, and Cole [15] also discussed the attitudes and intentions of elderly people who use staircase-assisted design at home. Among them, if the appearance or aesthetics of the related equipment is well-designed, the elderly will still regard it as a continuous stigma of disability due to the association of negative images. For example, poor social acceptance or loss of independence [16] will affect their willingness to use related equipment. It can be seen that the elderly want to look like ordinary elderly people but do not want to be regarded as dependent or disabled, which will affect their behavioral intentions and attitudes when using assistive devices.

Therefore, this research will focus on the factors that influence the behavior of middle-aged (45–64 years old) and elderly (65 years and older) people toward using walking aids in Taiwan. The subjects of the study were not people with severe walking disabilities, but people with mild walking disabilities or those who needed walking supporters because of old age. As Asians live in a closely connected social environment, middle-aged and elderly people are more concerned with social image. Based on the rational behavior theory, the relationship between “social image”, “attitude toward usage”, “subjective norm”, “safety level”, “usefulness”, and “behavior intention” will be verified by a survey investigation. According to the results of this study, it can provide a reference for government agencies and private manufactories to promote walking aids for middle-aged and elderly people.

## 2. Background and Hypotheses Development

### 2.1. Middle-Aged and Elderly People

Middle-aged citizens are considered to be those aged 45 to 65 years old, and the elderly those over 65 years old by the Middle-aged and Elderly Employment Promotion Act of the Ministry of Labor [17] in Taiwan. The Executive Yuan stated the purpose of formulating the regulations: “This middle-aged and senior citizens are mainly faced with age discrimination, decreased physical endurance, and social stereotypes” [18]. It can also be seen that the physical functions of middle-aged and elderly people gradually decline, which also leads to the decline of their social image, that is, the outside world tends to look down on their abilities. As of the end of last year (2019), the total population of middle-aged and older (45 years and older) was 10,681,705, accounting for 45.26% of the total population of the country (23,631,121). The phenomena bring the reality that many countries have to deal with them.

As the aging population increases, the proportion of people with reduced mobility is also increasing. Koon et al. [19] pointed out that decreased mobility is the most common disability condition among the elderly. Fifteen percent of the population over 65 years old (65–74 years old) have mobility problems and the population over 85 years old is as high as 48%. In addition, elderly people are prone to lower limb function weakness and poor balance due to illness or physical degradation, which may lead to falls [20]. Furthermore, the elderly rely on assistive devices to enhance their walking ability and improve the quality of life [21]. Therefore, even middle-aged people (45–64 years old), due to degeneration or disease, already face mobility problems. At the same time, the number of people with mobility impairments is increasing with age, and those with severe disabilities may even require assistance from others.

### 2.2. Design and Development of Walking Aids

In the Assistive Technology Act of the United States, assistive technology is divided into assistive technology equipment (devices) and assistive technology service (service) [17]. According to regulations in Taiwan, assistive technology equipment is the so-called assistive device, which refers to any product (including devices, equipment, instruments, technology, and software) [18]. The Bureau of Standards, Metrology, Inspection, and Quarantine of the Ministry of Economic Affairs in Taiwan formulated the classification standard CNS15390 for assistive devices in reference to the ISO 9999 specifications, which are divided into 11 categories such as personal medical assistive devices and personal mobility assistive devices. Wheelchairs, crutches, and walking aids are also personal mobility aids [20]. Others include one-arm operation walking aids, such as single crutches and four-legged crutches, and double-arm operation walking aids, such as axillary crutches, walkers, etc., as shown in Figure 1.

At present, in the design and research of walking aids, most of them are developed based on the consideration of biomechanics, functional aspect, or appearance [21,22,23,24]. In addition, Juan, Wang, and Sie [25] and others used the Situational Story Method or the Quality Function Deployment Method to explore the usage scenario and user’s needs. However, the psychological aspects of users or their attitudes toward the use of assistive devices are still rarely explored. It is also very important to understand the user’s willingness and use of the walking aids.

### 2.3. Social Image

Social image usually plays the role of motivation and cognition. It is a perception that will arouse and imitate real experience in the heart; the feedback obtained after a certain content is disseminated to the public [26]; and also a subjective cognition of the external objective world [27]. Cepeda Zorrilla, Hodgson, and Jopson [28] pointed out that social image results when people put forward different tastes or prestige views as a reaction to public opinion. Guo et al. believed that social image is an evaluation of the image of a certain behavior participant [29]. For walking aids users, social image is how people see them while using walking aids, which is different from seeing people who do not use them.

In addition, social image is a topic of concern in care-services-related research fields. Li and Wu [30] discussed the working environment and social image of care attendants by dividing social image into two aspects: “social status” and “professional license.” Moreover, Tsen [31] explored the benefits of promoting the social image of the mentally handicapped community, including social recognition, educational learning, and social care. Then, Hui-Fen [32] described some users’ reactions in the research of the elderly using assistive devices. Many elderly people think that the use of assistive devices will make them become disabled people and expressed other reactions, such as “I am not a patient, and why do I use these assistive devices?” The common thought among elderly people is that using assistive devices reveals a negative social image. Therefore, the elderly are unwilling to use assistive devices.

Salifu Yusif et al. reviewed academic journals from 1976 to 2015 and searched 39 articles from Google and Google scholar from 2000 to 2015 [10]. They found that among the barriers to the use of assistive technology by the elderly, the “Privacy” factor was most frequently mentioned (15 articles). “Stigma”, “Loss of Dignity”, “Lack of accessibility and social inclusion” and other factors were also related to the social image of the elderly as being unwilling to use assistive technology. From exploring the research and application of disadvantaged groups, it was found that social images are obviously related to the user’s attitude and intention of using assistive devices.

Since 2012, Dr. Chang has released conceptual designs of assistive devices that are based on a culturally creative and farsighted technology perspective, thinking about food, clothing, housing, transportation, education, and entertainment (shown in Figure 2) [33]. The design project emphasizes the need to provide basic health care and truly humane ways of responding to the needs of the elderly by considering their psychological aspects and providing aesthetic and practical designs, so that they can continue to live a decent and joyful life even in their twilight years. One of the works of the project, named “Puzzle for life” received the best work of the year. The concepts of the designs respond from the viewpoint of social image.

### 2.4. Theory of Reasoned Action

Theory of Reasoned Action (TRA) was proposed by Fishbein and Ajzen [34] to analyze how people change their behavior patterns through personal will. Human behavior is the result of rational thinking. The variables include attitudes, subjective norms, and behavioral intentions to explain and predict personal behavior. Attitude refers to the perception and evaluation of the implementation of specific behavior and also includes consideration of the subsequent results [35]. Subjective norms are the views and opinions held toward social pressure when engaging in specific behaviors. If other people hold opinions for (or against) something, the easier (or harder) it is to execute the act [36]. Furthermore, behavioral intention is the willingness to engage in a specific behavior, and it is a prerequisite for actual execution [37]. It will be affected by two elements: attitude and subjective norms.

Concerning health care and medicine-related fields, for example, the study of organ donation after death [38] explored the attitude and intention of organ donation through rational behavior theory and explained the rational decision-making process of this behavior. Moreover, in the study of the condom use behavior of people living with HIV [39] it was found that the subjective norms of condom use in individual cases can significantly predict the possibility of condom use in future sex. TRA is also used to explain how to improve health and develop good habits by increasing healthy behaviors and reducing habitual behaviors [40], for example, teachers’ use of interventions to develop student habits to reduce the sitting time in the classroom [41] or related research on pregnant women’s exercise behavior [42]. Rational behavior theory is also widely used in the study of the behavioral intentions of healthy people, patients, and middle-aged and elderly people.

Regarding empirical research on social image and usage attitude, Lin and Bhattacherjee [43] discussed the use of online entertainment systems. The research result showed that perceived entertainment and social image positively affect usage attitude. Therefore, social image will have a positive impact on the user’s attitude.

In short, for middle-aged and elderly people, the perception of the social image of using a walker will have a positive impact on their attitude toward usage. Similarly, changes in attitudes and subjective norms will also affect the behavioral intention of using walking aids. Therefore, this research proposes the following hypotheses:

**Hypothesis** **1.***Social image will have a positive and significant influence on the attitude toward using walking aids*.

**Hypothesis** **2.***The attitude toward using walking aids will have a positive and significant influence on behavioral intention*.

**Hypothesis** **3.***Subjective norms will have a positive and significant influence on the behavioral intention of using walking aids*.

### 2.5. Safety

The safety aspect refers to safety awareness, which means whether people can be aware of their own safety [44]. The higher the awareness of safety, the more care and attention is paid to health or life [45]. In addition, the user’s perception of safety is a cognitive process that affects emotions and behavioral intentions. It is expressed through the interaction between cognition, emotion, and behavioral intention. The need for safety means the importance of and attitude to safety, which are also proportional to the intention of use [46,47]. The safety level of walking aids refers to the stability and support of the walking aids [22,24], that they will not cause falls and secondary injuries [21].

In addition, safety needs to be considered in the research on intelligent nursing aids. For example, Kwee-Meier, Bützler, and Schlick [48] found in the study of wearable positioning systems that perceptual safety performance has a positive and significant impact on behavioral intentions. Consideration of the potential safety risks of smart home devices also is an important factor in whether consumers are willing to use smart devices [49]. In addition, research on intelligent nursing services for the elderly found that due to uncertain factors, especially perceived safety, the elderly are reluctant to use the intelligent call system. If the elderly can feel the degree of safety, the intention of use will be more obvious [50].

All in all, the perceived security level has a positive effect on behavioral intentions. For the elderly, the higher the perceived safety, the higher their intention to use walking aids. Therefore, this research infers that the perception of the middle-aged and elderly of the safety level has a positive and significant impact on behavioral intentions. The proposed hypothesis is:

**Hypothesis** **4.***The perceived safety level of walking aids will positively and significantly affect the behavioral intention of using walking aids*.

### 2.6. Usefulness

Fred Davis [51] proposed the Technology Acceptance Model (TAM), which pointed out that through perceived usefulness and perceived ease of use, as well as other variables, it is possible to predict whether users can accept a particular technology. Perceived usefulness means that when a certain technology can improve work performance, the willingness to use it will be greatly increased, and the work performance will also be improved accordingly. In addition, if users believe that related services are beneficial to their daily life or work, they will increase their intentions; if they find them easy to use, they will also increase their overall willingness to use them [52].

There are many related studies. For example, Lin and Chang [53] studied the factors that affect the willingness of the elderly to use, such as production and sales systems. It was found that subjective norms, the visibility of pictures and labels all have a significant impact on perceived usefulness. In addition, when discussing the opinions of the elderly on the practicality and their attitudes toward autonomous vehicles, the acceptance was very high [53]. Therefore, it can be stated that usefulness has a significant positive impact on usage intention and attitude.

In summary, usefulness has a positive effect on behavioral intentions. For users of walking aids, the higher the perceived usefulness, the higher will be their behavioral intention to use the aids. Therefore, the proposed hypothesis is:

**Hypothesis** **5.***The usefulness of walking aids will positively and significantly affect the user’s behavioral intentions*.

### 2.7. Mediation Effect

In the applied research on social image and usage attitude, Lin and Schlick [44] verified the positive correlation between social image and usage attitude. Attitudes positively influence behavioral intentions. Moreover, researches on the behavior of the different genders on the recycling of resources on campus and the behavioral research on information acceptance by the elderly have found that there is a direct correlation between use attitude and behavioral intention [54,55].

Based on the above research, social image has a positive influence on usage attitude, and usage attitude has a positive influence on behavior intention. For the user of walking aids, the more positive the social image of using walking aids is, the more positive the attitude toward using aids will be. Therefore, this study infers that middle-aged and elderly people’s social image perception of using walking aids will have a positive and significant impact on behavioral intentions through their attitudes. The proposed hypothesis is:

**Hypothesis** **6.***The attitude toward the usage of walking aids has a mediation effect between social image and behavioral intention*.

This research explores the behavioral intentions of the middle-aged and the elderly toward the use of walking aids and studies the influencing factors. This study took social image, usage attitude, subjective norms, safety level, and usefulness as independent variables. Among them, usage attitude also played the role of a mediating variable. Behavioral intention was the dependent variable. Through the literature review, the research model is drawn as shown in Figure 3.

## 3. Materials and Methods

### 3.1. Data Collection

This study explored the intention of middle-aged and elderly people to use walking aids, so the research subjects were people over 45 years old. This research used questionnaires, which were conducted online and in the form of paper, which was assisted by volunteers. In the north, middle, south, and east of Taiwan, we collected research data through community elderly associations, religious groups, civil organizations, and assistive device manufacturers. The questionnaire collection period of time was from 1 May 2020 to 30 June 2020. After deleting invalid questionnaires, there were a total of 406 valid questionnaires. According to the sample size requirement formula from the survey system website [56], when the statistical confidence was 95% and the confidence interval was 5%, the sample size of the total population of Taiwan was 384 copies. Therefore, the sample size of this research was sufficient.

### 3.2. Measurement Instrument

The research questionnaire included the investigation of the basic participants’ background information and their opinions of the research constructs. The background information, including gender, age, education, marital status, occupation, family environment, and residential area, was investigated in the first part of the questionnaire. Then, in the second part of the questionnaire the six latent variables were measured by a Likert seven-point scale ranging from “strongly disagree” (1) to “strongly agree” (7). The measurement items were mainly adapted from the previous studies. All items were originally written in Chinese and modified to the suitable walking aids usage scenario for the survey participants. First, there were five items related to users’ attitudes toward using walking aids. Three items were adapted from Wu and Lin’s study [57], one item was from Chen and Zheng’s study [58], and the other one item was from Zhu, Huang, and Weng’s work [59]. Then, for the subjective norm construct, all five items were adapted and modified from Zhang and Sink’s study [60], which concerned people’s walking activities in Taipei. The research target and scenario were similar to the present study. It contained encouragement from several aspects, such as families, friends, the elderly, the government, and environmental groups. After that, there were also five items about safety consideration. One item came from Lai’s research [61] in the safety design of a public bicycle. Two items were adapted from Zhou and Lee’s study [62] about the safety level of club members. The other two items were modified from Zhuang and Huang’s study [63] about the well-being of native elementary students while walking on the community roads. Then, usefulness was also evaluated by five items. The first two items were adapted from Zeng, Hong, and Lin’s research [64]. Their research concerned using technology to solve problems. The other three items were from Xu et al.’s study, which concerned how a blog can provide useful information [65]. Moreover, three items about social image measurement were taken from Yang et al.’s research providing social image evaluation items for wearable devices [66]. The other two items came from Lin and Bhattacherjee’s study [43], which was originally designed for online video game users about their perceived social image. Then, for the behavioral intention of using walking aids, the first two items were adapted from Lee and Gu’s research [67] about the behavioral intention of purchasing souvenirs abroad. The other two items came from Chen et al.’s study, which explored the purchasing information and recommendation effects on purchasing intention [68]. The last item was adapted from Zhang et al.’s study about the usage of the voice message board [69].

### 3.3. Data Analysis

The results were processed in three sections: descriptive statistical analysis, measurement model verification, and structural equation modeling analysis. In this study, the structural equation model (SEM) was used to obtain the relevant conclusions. The Likert seven-point scale used in the questionnaire was a continuous scale, which is statistically consistent with a normal distribution and obtains relatively accurate results. However, some of the research variables in this study were nominal scales, such as gender, marriage, or occupation. These variables could not be analyzed using structural equations, which limit the results of the research [70,71,72]. Descriptive statistical analysis by SPSS contained two parts: one was the frequency distribution calculation of demographic data for a basic understanding of the sample set; and the other one was the mean value and standard deviation of the items of each construct. Then, according to Anderson and Gerbing [73], the study provided verification of the measurement model by confirmatory reliability analysis, convergent validity, and discriminant validity. After that, based on the research model, the structural equation model was analyzed, including path analysis and mediation effect analysis by statistic software AMOS (SPSS Inc., Chicago, IL, USA).

## 4. Results

### 4.1. Descriptive Statistical Analysis

#### 4.1.1. Frequency Distribution

The categorical data elements of the 406 valid questionnaires included gender, age, education, marital status, occupation, family environment, and residential area. The respondents were 70.2% female. Most of respondents were aged 45–64, which was the definition of middle-aged people. A total of 45.32% of the respondents were aged between 45 and 54, and 39.66% of respondents were aged between 55 and 64. Regarding the education level, the largest group was college and university, which was 47.04%. A total of 361 respondents were married, which was 88.92%. The data are shown in Table 1.

#### 4.1.2. Item Statistical Analysis

Table 2 shows the mean values and standard deviation values of items of each construct. The lowest average score of all items was 4.57, which was for “I can get the envy of my friends by using advanced walking aids” of the social image construct. The highest average score was 6.37, for “If I use walking aids well, I will convey positive news to others” of the behavioral intention. This time also had the highest standard deviation value of all items. There were two tied lowest standard deviation values in the behavioral intention construct.

The F-test of different ages in the social image construct reached a significant level (F = 10.676, *p* < 0.05). It meant that age made a significant difference in the social image construct. After comparing with the Scheffe method, it was found that subjects 55–64 years old had a significantly higher opinion in the social image construct than those 45–54 years old, and subjects 65 years or above had a significantly higher opinion in social image than those 45–54 years old. The testing results are shown in Table 3.

### 4.2. Measurement Model Verification

#### 4.2.1. Convergent Validity

A complete structural equation model should be implemented by two phases; the first phase is to verify the measurement model and then the second phase is the structure mode analysis [73]. Confirmatory factor analysis provides standardized factor loading, composite reliability, and average variance extracted to test the convergent validity. After the acceptance of convergent validity verification, the next phase of structure model can be processed [74]. The standardized factor loading should be larger than 0.6. It is better that the value is larger than 0.7 [75]. If the loading value is less than 0.45, the item should be deleted [76]. Then, the suggested value for composite reliability should be larger than 0.6, and the average variance extracted value should be larger than 0.5 [77,78,79].

Table 4 shows the results of the confirmatory factor analysis. All standardized factor loadings were between 0.634 and 0.956, larger than 0.6, which meant all items had acceptable reliability. All composite reliability values were between 0.875 and 0.964, larger than 0.7, which meant all constructs had acceptable reliability. Finally, all average variance extracted values were between 0.589 and 0.843, larger than 0.5, which meant all constructs had acceptable convergence validity.

#### 4.2.2. Discriminant Validity

This study used average variance extracted as the criteria for testing the discriminant validity of research constructs. According to Fornell and Larcker [77], the square root of the average variance extracted (AVE) value (the bold figures in Table 5) should be larger than the Pearson correlative coefficients with other constructs (the figures under the diagonal) in order to be discriminant with each other. Table 5 shows the discriminant validity test of the measurement model. The values under the diagonal are the Pearson correlation coefficients between constructs, which were smaller than the square root of average variance extracted values on the diagonal. Therefore, all the discriminant validity of research constructs were acceptable.

### 4.3. Structural Equation Model

#### 4.3.1. Structural Model Analysis

Schumacker and Lomax [80] and Kline [74] pointed out that, since the big sample analysis would cause a *p*-value smaller than 0.05, the model fit would be affected as a bad result. Therefore, the quantitative research should adapt several different methods to test the model fit. The present study implemented eight common models of fit verification methods proposed by Jackson, Gillaspy, and Purc-Stephenson [81]. Moreover, if the sample size was big, larger than 200, the Chi-square value easily got a bad result. The bootstrap method provides an alternative way to get a better result [82]. By using Chi-square divided by degree of freedom (DF), the ideal result should be less than three. Moreover, other criteria provide a more rigorous standard for model fit verification, as Table 6 shows. For example, the Root Mean Square Error of Approximation (RMSEA) value should be smaller than 0.08 [83]. Comparative Fit Index (CFI) criteria should be larger than 0.9. The tested results are shown in Table 6. All the model fit criteria tested fitted the suggested standards [80].

#### 4.3.2. Path Analysis

Table 7 shows the path coefficient analysis for verification of the causal relationship between variables. As the results show, social image positively impacted the attitude toward usage of walking aids significantly. The unstandardized regression coefficient from social image to attitude toward usage was 0.243. The *p*-value was less than 0.001, which meant social image impacted attitude toward usage significantly. The explainable variation was 0.206. The coefficients from attitude toward usage, subjective norm, safety, and usefulness to behavioral intention, were 0.246, 0.208, 0.283, and 0.178, respectively. Their *p*-values also were less than 0.001, except usefulness. The explainable variation was 0.495. Figure 4 shows the regression coefficients of the structural equation model.

#### 4.3.3. Mediation Effect Analysis

This study used bootstrapping as the repeated sampling method to produce statistical confidence intervals for indirect effects. Table 8 shows the results of social image impacting behavioral intention through attitude toward usage. The confidence intervals were 0.003 to 0.113, and the *p*-value was less than 0.05. The value of lower bound and upper bound was not across 0, which meant the indirect effect existed.

## 5. Discussion

This research was mainly to investigate the behavioral intention of middle-aged and elderly people to use walking aids and to explore its influencing factors. Based on the Theory of Reasoned Action (TRA), combined with the social image and attitude toward usage as mediating variables, the research framework and related hypotheses were proposed. After collecting data through a questionnaire survey, the structural equation model was used to test the model and verify related hypotheses. Based on the Theory of Reasoned Action (TRA), this research attempted to explore the factors affecting the behavior of middle-aged and elderly users from the perspective of social image. Therefore, integrating factors such as safety, usefulness, and social image dimensions were used to discuss the influence on middle-aged and elderly people’s intention to use walking aids. The research results are as follows.

### 5.1. The Influence of Social Image on the Attitude toward Using Walking Aids

The results of the study showed that social image has a significant positive impact on attitudes toward using walking aids. This result was consistent with the previous research [43]. The average score of social image measurement was 4.876, which was the lowest score of all constructs. It could be that the middle-aged and elderly people think that the use of walking aids cannot be socially recognized. They are obviously worried about the external image of using a walker, so they have reservations about using a walker, and even have a sense of resistance. At the same time, the standard deviation of the social image item “I use walking aids to make a good impression” was greater than 1.3, indicating that users’ opinions varied greatly. Moreover, the item “I can get the envy of my friends by using advanced walking aids” had the lowest mean score, 4.57, and the highest standard deviation, 1.5. The low score may show that on the issue of the use of walking aids, the users thought that even choosing high-end-style walking aids would not necessarily reverse the social image. Therefore, they could not gain support and recognition for using walking aids. At the same time, the high standard deviation may indicate that users had greatly inconsistent opinions, and it also may show that the public has unequal attitudes and opinions toward people who use walking aids.

In the path analysis of the influence of social image on use attitude, the non-standardized regression coefficient reached 0.243, indicating that social image factors had a high proportion of influence on use attitude. It also confirms the hypothesis of this study that social image has a significant influence on the attitude toward using walking aids. Therefore, it is also proved that middle-aged and elderly users are obviously concerned about the social impression of disability, aging, and the dependence caused by the use of walking aids. This differs from ordinary people, so it affects their attitude toward using mobility aids. At the same time, if they were to choose a more advanced walker, whether it was for a change in function or appearance, it would not enhance the attitude toward using a walker.

### 5.2. The Influence of Attitude and Subjective Norms on the Behavioral Intention of Walking Aids

The results of this study showed that the attitude toward using walking aids has a significant positive effect on behavioral intentions. This result was similar to the results of previous studies [55]. In addition, subjective norms have a significant positive effect on the behavioral intention of using walking aids. This result was also consistent with the results of previous studies [39,84], and all fit into the framework of the TRA.

In the attitude toward usage construct, a positive view appeared, which meant that users agree that if their walking function was affected in the future, they would be open and willing to accept using walkers. The behavioral intention construct reached the highest average score of 6.324, indicating that if having difficulty walking in the future, the participants would use walking aids to assist walking, and even to convey positive messages or to recommend it to others. The subjective norm item “I will try to use it because the professional recommended the benefits of walking aids” gained the highest average score, 6.36, and its standard deviation value was 0.77. That means not only high agreement among the participants on this question, but also that as long as the use of walking aids is recommended by professionals, the user’s willingness to use it will be greatly increased.

### 5.3. The Impact of Safety and Usefulness on the Behavioral Intention of Using Walking Aids

The intensity of the perception of safety has a significant impact on the behavioral intention of using walking aids. This result was similar to the results of previous studies [46,47]. In addition, the usefulness of walking aids also has a significant positive impact on behavioral intentions. This result was also similar to the results of previous studies [19,85].

For users, the average score of the safety aspect was 5.838, which meant that they are less worried about the safety of walking aids. At the same time, in the safety construct, the non-standardized regression coefficient of behavioral intention reached 0.283. In addition to being an important influencing factor on the behavioral intention of using walking aids, it also meant that after users feel that safety is guaranteed, their willingness to use walking aids will increase significantly. In addition, the score of related items in usefulness also reached 6.01, and standard deviation was 0.85, indicating that users agree that walking aids are useful. However, since the regression coefficient was 0.178, the lowest of all factors, the degree it affected the behavioral intention of using walking aids was not so obvious.

### 5.4. Social Image Influences Behavioral Intention through Attitude toward Usage

Social image affects the attitude toward usage, and the attitude toward usage significantly positively affects the behavior intention, that is, the social image of using walking aids indirectly affects the behavioral intention. The attitude toward usage had a mediating effect, and this result was similar to the previous studies [19,42]. It can be seen that if middle-aged and elderly people are more concerned about their social image, it will affect their attitudes to using walking aids, which, in turn, will affect their behavioral intention to use walking aids.

## 6. Conclusions

The World Health Organization (WHO) proposed the Global Aging and Health Strategy and Action Plan in 2016 [3], which mentioned that the physical and mental functions of the elderly are rapidly decreasing. The impact of the elderly on the whole of society is very important. If the elderly can maintain their functional capacity through their interaction with the environment, and be provided with adequate support for improving their health, then the life span of the elderly can be extended. The elderly can participate in society and improve their own health and well-being. Therefore, the use of assistive equipment and a barrier-free environment are the best supports for maintaining the functions of the elderly.

Taiwan began to implement National Ten-year Long-term Care Plan in 2008, providing assistive equipment and barrier-free environment improvement subsidies, including crutches, walking aids, or walkers. In 2018, the National Ten-year Long-term Care Plan 2.0 was promoted [86], and the “Reablement” service was updated. Professional medical staff provide daily life function training, physical function training and maintenance, assistive devices use training, and other services. The goals of the program include improving the individual’s ability to live independently, achieving ability recovery, and increasing the ability to act independently.

As many elderly people get older, the function of their lower limbs deteriorates, which affects their walking function. However, they are afraid of being laughed at, their external image becoming morbid or weak. This may be why the middle-aged and elderly are reluctant to use walking aids. Instead, they will walk with umbrellas or bamboo poles, which is not only very laborious but also prone to danger. Therefore, it is very important to understand the influence of “social image” on users for the research on the use of walking aids by middle-aged and elderly people. In the past, research on assistive devices mostly focused on the design of the mechanics, functional aspects, and appearance of walking assistive devices. The behavior of assistive devices from the perspective of the user’s mentality and the relationship between related influencing variables were rarely studied.

Therefore, improving life functions through assistive devices is very important in an aging society. According to the present study, for these middle-aged or elderly people who are facing gradual degradation of functions, if the use of assistive devices cannot overcome the existing poor social image, no matter how sophisticated the assistive device is, it still cannot increase the willingness to use it. The group between the ages of 45–55, especially, was more concerned with social image than other elderly groups. It also becomes impossible to use assistive devices to achieve social participation, thereby delaying aging and increasing physical and mental health.

This study mainly sampled the middle-aged and elderly people in the urban area. The results of the study indicated that the sampled subjects had relatively high academic qualifications; a college degree or above accounted for 74.4%. However, the higher the level of education, the higher the ability to absorb information about assistive devices, and the higher the proportion of self-help ability in understanding the use of assistive devices. This study may not be able to fully present the attitudes and intentions of other people with lower academic qualifications. In addition, the majority of the subjects were under 65 years old (85%), and the elderly group sample (65 years old or above) was relatively small. Future research can improve in the sampling process. In addition, this research focuses on walking aids. In the future, research can focus on the intention of using other types of assistive devices, such as wheelchairs, hearing aids, or visual-impairment-related assistive devices, providing more about the use of various assistive devices.

## Figures and Tables

**Figure 1 healthcare-08-00543-f001:**
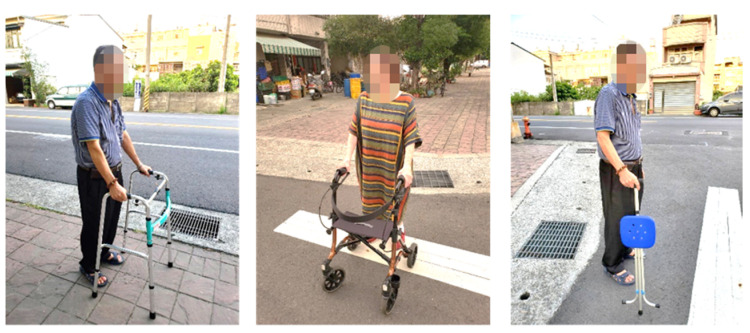
Walking aids (the images were shot by the authors).

**Figure 2 healthcare-08-00543-f002:**
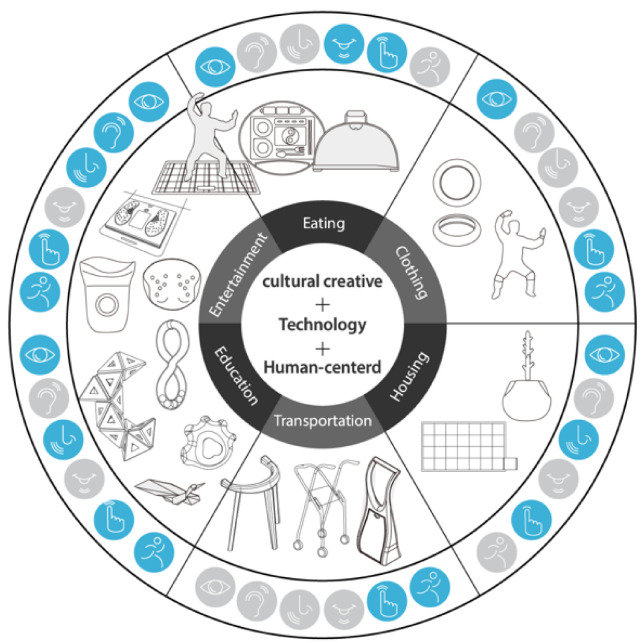
Conceptual design of assistive devices.

**Figure 3 healthcare-08-00543-f003:**
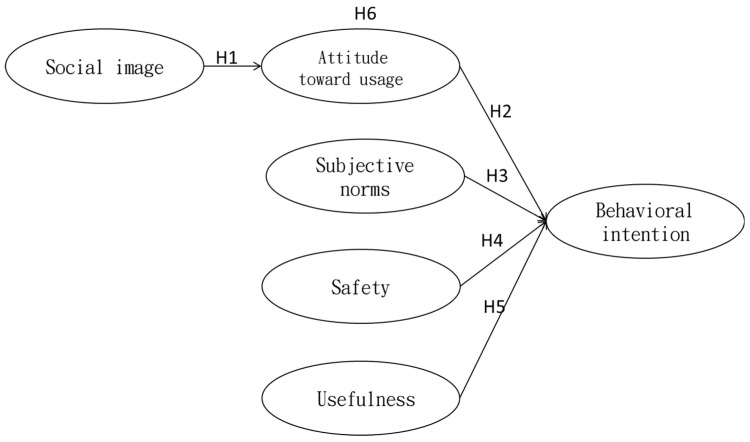
Research Model.

**Figure 4 healthcare-08-00543-f004:**
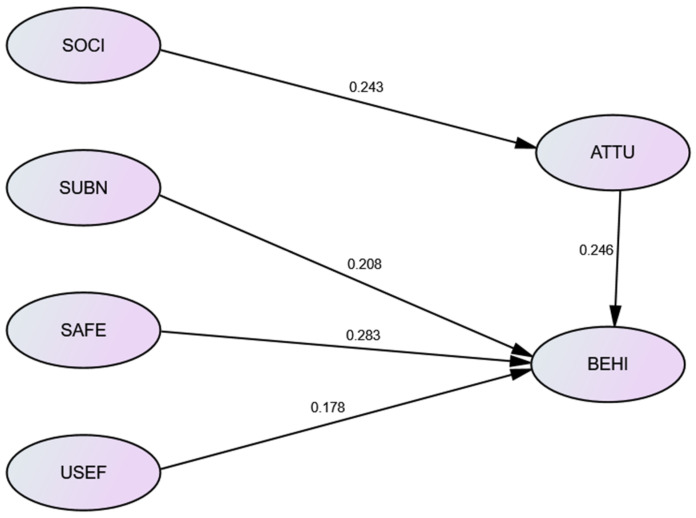
SEM statistic model.

**Table 1 healthcare-08-00543-t001:** Frequency distribution table.

Variable	Value Label	Frequency	Percent	Accumulated Percent
Gender	Male	121	29.80	29.80
	Female	285	70.20	100.00
	Total	406	100.0	
Age	45–54 years old	184	45.32	45.32
	55–64 years old	161	39.66	84.98
	65 years or above	61	15.02	100.00
	Total	406	100.0	
Education	Elementary	4	0.99	0.99
	Junior school	6	1.48	2.46
	High school/vocational school	94	23.15	25.62
	College/University	191	47.04	72.66
	Master or above	111	27.34	100.00
	Total	406	100.0	
Marital status	Married	361	88.92	88.92
	Unmarried	45	11.08	100.00
	Total	406	100.0	
Occupation (Background)	Medical	32	7.88	7.88
	Agriculture, Forestry and Fisheries	10	2.46	10.34
	Military service/civil servants/teachers	96	23.65	33.99
	Service industry	93	22.91	56.90
	Manufacture/business	61	15.02	71.92
	Freelance	41	10.10	82.02
	Housekeeping	54	13.30	95.32
	Others	19	4.68	100.00
	Total	406	100.0	
Family environment	Bungalow	13	3.20	3.20
	Townhouse	236	58.13	61.33
	Apartment (no elevator)	40	9.85	71.18
	Building (with elevator)	117	28.82	100.00
	Total	406	100.0	
Residential area	Suburbs	45	11.08	11.08
	Urban area	229	56.40	67.49
	Countryside	42	10.34	77.83
	Small town	75	18.47	96.31
	Medium-sized city	15	3.69	100.00
	Total	406	100.0	

**Table 2 healthcare-08-00543-t002:** Mean and standard deviation of items.

Construct	Item	Mean	SD ^1^
Social image	1. I use walking aids to make a good impression.	4.98	1.31
	2. Using walking aids can enhance my image among community residents.	4.85	1.32
	3. I feel that using walking aids brings me recognition from others.	5.01	1.36
	4. I will get more respect from others when I walk with walking aids.	4.97	1.40
	5. I can get the envy of my friends by using advanced walking aids.	4.57	1.50
Total		4.876	
Attitude toward usage	1. I think the use of walking aids will meet my requirements.	5.83	1.12
	2. I think using walking aids will be a pleasant experience.	5.37	1.30
	3. Overall, my evaluation of the use of walking aids is positive.	6.10	0.91
	4. I think using walking aids is a good choice.	6.07	0.93
	5. I think the use of walking aids is really needed by the elderly.	6.10	1.18
Total		5.894	
Subjective norm	1. I will use it because my family members (including my spouse) persuaded the use of walking aids to be healthy.	6.17	0.81
	2. I will try to use it because my friends agree with the use of walking aids.	5.96	1.00
	3. I will try to use it because my neighbors persuaded the advantage of walking aids.	5.78	1.09
	4. I will use walking aids as much as possible because of government policies.	5.77	1.10
	5. I will try to use it because the professional recommended the benefits of walking aids.	6.36	0.77
Total		6.008	
Safety	1. The design of the walking aid itself is extremely safe.	6.00	0.85
	2. I think walking aids are trustworthy.	5.94	0.85
	3. Using walking aids always makes people feel safe.	5.93	0.90
	4. I think it is safe to walk on the street with a walking aid alone.	5.50	1.18
	5. I think it is safe to use walking aids to walk in the community.	5.82	1.00
Total		5.838	
Usefulness	1. I think using walking aids can help achieve the goal of independent walking.	6.10	0.80
	2. Using walking aids will solve my walking problems.	5.97	0.90
	3. Using walking aids will help me walk more safely.	6.11	0.83
	4. Using walking aids will increase my walking speed.	5.71	1.07
	5. Using walking aids will help me effectively prevent falls.	6.16	0.85
Total		6.010	
Behavioral intention	1. If I have difficulty walking, I plan to use a walker.	6.28	0.76
	2. If I have difficulty walking, I will be willing to use walking aids.	6.32	0.72
	3. If I use walking aids well, I would recommend them to others.	6.34	0.76
	4. If I use walking aids well, I will convey positive news to others.	6.37	0.72
	5. If I have difficulty walking, I will often use walking aids.	6.31	0.73
Total		6.324	

^1^ SD = Standard Deviation.

**Table 3 healthcare-08-00543-t003:** Age vs. Social image research construct analysis of variance table.

Construct	Age	N	Mean	Std. Deviation	F-Test	*p*-Value Sig.	Scheffe
Social image	1. 45–54 years old	184	4.834	1.163	10.676	*p* < 0.001	2 > 1, 3 > 1
	2. 55–64 years old	161	5.201	1.061			
	3. 65 years or above	61	5.525	0.958			
	Total	406	5.083	1.120			

N = Number; Std. = Standard; Sig. = Significant.

**Table 4 healthcare-08-00543-t004:** Confirmatory factor analysis.

Construct	Ite	Significance of Estimated Parameters				Item Reliability		Composite Reliability	Convergence Validity
		Unstd. ^1^	S.E.	Unstd./S.E.	*p*-Value	Std. ^2^	SMC ^3^	CR ^4^	AVE ^5^
Social image	SOCI1	1.000				0.900	0.810	0.951	0.796
	SOCI2	1.046	0.033	31.284	0.000	0.930	0.865		
	SOCI3	1.041	0.035	29.774	0.000	0.917	0.841		
	SOCI4	1.060	0.036	29.508	0.000	0.917	0.841		
	SOCI5	1.003	0.047	21.455	0.000	0.791	0.626		
Attitude toward usage	ATTU1	1.000				0.634	0.402	0.875	0.589
	ATTU2	1.255	0.105	11.999	0.000	0.672	0.452		
	ATTU3	1.105	0.077	14.394	0.000	0.898	0.806		
	ATTU4	1.130	0.077	14.595	0.000	0.918	0.843		
	ATTU5	0.918	0.079	11.586	0.000	0.666	0.444		
Subjective norm	SUBN1	1.000				0.841	0.707	0.922	0.705
	SUBN2	1.259	0.051	24.590	0.000	0.921	0.848		
	SUBN3	1.300	0.059	21.940	0.000	0.870	0.757		
	SUBN4	1.259	0.063	20.024	0.000	0.820	0.672		
	SUBN5	0.843	0.048	17.414	0.000	0.735	0.540		
Safety	SAFE1	1.000				0.874	0.764	0.931	0.730
	SAFE2	1.001	0.037	26.713	0.000	0.905	0.819		
	SAFE3	1.014	0.038	26.346	0.000	0.903	0.815		
	SAFE4	1.147	0.059	19.471	0.000	0.777	0.604		
	SAFE5	1.009	0.049	20.670	0.000	0.805	0.648		
Usefulness	USEF1	1.000				0.793	0.629	0.918	0.694
	USEF2	1.268	0.061	20.920	0.000	0.892	0.796		
	USEF3	1.201	0.056	21.459	0.000	0.913	0.834		
	USEF4	1.268	0.079	15.975	0.000	0.730	0.533		
	USEF5	1.128	0.061	18.592	0.000	0.823	0.677		
Behavioral intention	BEHI1	1.000				0.934	0.872	0.964	0.843
	BEHI2	0.991	0.024	40.596	0.000	0.956	0.914		
	BEHI3	0.926	0.032	28.941	0.000	0.881	0.776		
	BEHI4	0.898	0.030	30.339	0.000	0.897	0.805		
	BEHI5	0.954	0.028	34.019	0.000	0.921	0.848		

^1^ Unstd. = Unstandardized factor loading, ^2^ Std. = Standardized factor loading, ^3^ SMC = Squared Multiple Correlations, ^4^ CR = Composite reliability, ^5^ AVE = Average Variance Extracted.

**Table 5 healthcare-08-00543-t005:** The result of discriminant validity analysis.

	AVE	SOCI	ATTU	SUBN	SAFE	USEF	BEHI
SOCI ^1^	0.796	**0.892**					
ATTU ^2^	0.589	0.454	**0.767**				
SUBN ^3^	0.705	0.515	0.234	**0.84**			
SAFE ^4^	0.730	0.590	0.268	0.740	**0.854**		
USEF ^5^	0.694	0.571	0.259	0.630	0.693	**0.833**	
BEHI ^6^	0.843	0.481	0.399	0.587	0.632	0.557	**0.918**

^1^ SOCI = Social Image,^2^ ATTU = Attitude toward Usage, ^3^ SUBN = Subjective Norm, ^4^ SAFE = Safety, ^5^ USEF = Usefulness, ^6^ BEHI = Behavioral Intention.

**Table 6 healthcare-08-00543-t006:** Model fit verification.

Model Fit	Criteria	Model Fit of Research Model
χ^2 1^	The smaller the better	522.511
DF ^2^	The larger the better	394.000
Normed Chi-square (χ^2^/DF)	1 < χ^2^/DF < 3	1.326
RMSEA ^3^	<0.08	0.028
TLI (NNFI) ^4^	>0.9	0.989
CFI ^5^	>0.9	0.990
GFI ^6^	>0.9	0.960
AGFI ^7^	>0.9	0.952

^1^ χ^2^ = Chi-square, ^2^ DF = Degree of Freedom, ^3^ RMSEA = Root Mean Square Error of Approximation, ^4^ TLI (NNFI) = Tucker-Lewis Index (Non Normed Fit Index), ^5^ CFI = Comparative Fit Index, ^6^ GFI = Goodness of Fit Index, ^7^ AGFI = Adjusted Goodness of Fit Index.

**Table 7 healthcare-08-00543-t007:** Path analysis.

DV ^7^	IV ^8^	Unstd. ^9^	S.E. ^10^	Unstd./S.E.	*p*-Value	Std. ^11^	R2 ^12^
ATTU	SOCI ^1^	0.243	0.031	7.889	*p* < 0.001	0.454	0.206
BEHI ^6^	ATTU ^2^	0.246	0.068	3.616	*p* < 0.001	0.228	0.495
	SUBN ^3^	0.208	0.069	2.992	0.003	0.206	
	SAFE ^4^	0.283	0.062	4.535	*p* < 0.001	0.315	
	USEF ^5^	0.178	0.073	2.426	0.015	0.151	

^1^ SOCI = Social Image, ^2^ ATTU = Attitude toward Usage, ^3^ SUBN = Subjective Norm, ^4^ SAFE = Safety, ^5^ USEF = Usefulness, ^6^ BEHI = Behavioral Intention, ^7^ DV = Dependent Variable, ^8^ IV = Independent Variable, ^9^ Unstd. = Unstandardized regression coefficients, ^10^ S.E. = Standard Error, ^11^ Std. = Standardized regression coefficients, ^12^ R2 = Explainable variations.

**Table 8 healthcare-08-00543-t008:** Indirect effect analysis.

Indirect Effect	Point Estimate	Product of Coefficients			Bootstrap 1000 Times	
					Bias-Corrected 95%	
		S.E.	Z-Value	*p*-Value	Lower Bound	UPPER BOUND
SOCI ^1^→ATTU ^2^→BEHI ^3^	0.060	0.027	2.215	0.027	0.003	0.113

^1^ SOCI = Social Image, ^2^ ATTU = Attitude toward Usage, ^3^ BEHI = Behavioral Intention.

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
