# Peer review of "Social Image Impacting Attitudes of Middle-Aged and Elderly People toward the Usage of Walking Aids: An Empirical Investigation in Taiwan"

_healthcare, 2020, doi:10.3390/healthcare8040543_

Round 1

Reviewer 1 Report

Review of “Social image impacting attitudes of middle-aged and elderly people toward the usage of walking aids: An empirical investigation in Taiwan”

Starting from Theory of Reasoned Action (TRA), the research aims to investigate how social image can affect attitudes of middle-aged and elderly people toward the usage of walking aids. Specifically, the paper poses various hypotheses that are tested exploiting 457 questionnaires filled in by Taiwanese patients.

The study face an interesting issue, from both a scientific and practical viewpoint, and it is carried out correctly in my opinion. However, some improvements to the paper can be suggested as reported by the list below:

  • The introduction should be more oriented on behavioural intentions and attitudes as they are the most important aspect investigated by the study. The relevance of elderly now and in future is well underlined, while the importance of such behavioural/social aspects should be better emphasize in the introduction of the work.
  • One other important aspect, I think the author should improve is the usage of surveys for this kind of research. I think you adequately use a survey mean for this study. However, I would like to see a discussion of the potential limits related to such tools in this kind of research. This aspect can be faced in “materials and method” chapter or in the “Research limitations” (personally I prefer in the methodological section). For underling such aspect, I suggest to refer to past studies, and at least at: -Stefanini, A., Aloini, D., Gloor, P., & Pochiero, F., (2020); Patient Satisfaction in Emergency Department: Unveiling Complex Interactions by Wearable Sensors; Journal of Business Research. Article in press. DOI: 10.1016/j.jbusres.2019.12.038.   - Carayon, P., Kianfar, S., Li, Y., Xie, A., Alyousef, B. and Wooldridge, A. (2015), “A systematic review of mixed methods research on human factors and ergonomics in health care”, Applied Ergonomics, Vol. 51, pp. 291-321.    - Stefanini, A., Aloini, D., & Gloor, P. (2020); Silence is golden: the role of team coordination in health operations; International Journal of Operations and Production Management. Article in press. DOI: 10.1108/IJOPM-12-2019-0792.
  • In my opinion, the sub-chapter 1.1, 1.2, 1.3, 1.4, 1.5, 1.6, 1.7 should be included in a new chapter (and not in the introduction section), which can be called “Background and Hypotheses development”.
  • In the 4.2 section “practical implications”, it is unclear/misleading the presence of “1. Health and medical policy advocacy”; “2.Education turns the social image of using walking aids”; “3.Precautions for the design and sales of walking aids” in the middle of the text. I understood that you used it for introducing different aspects, but it is not good way to do that in my opinion (they seems like sub-chapters).

Reviewer 2 Report

Thank you for your contributions to this important area of physical and mental health.  Just a couple comments/recommendations for revision:

1.  Line 100 includes the term "Orientals."  Consider changing to "Asians."  However, I see the authors are based in Taiwan and likely know the most culturally sensitive and appropriate terms.  Please disregard if my comment is not seen as necessary in Taiwanese culture.

2.  With the small sample of elderly population, is it necessary to include them in the analysis?  It seems the middle-aged population would likely face different social pressures compared to the elderly (middle-aged people needing walking assistance are likely outnumbered socially -- they are likely surrounded by peers who do not need walking devices).  This social pressure seems different from the elderly population, who is likely surrounded by peers who also do need assistance.  With this discrepancy, would the data be even cleaner to remove the small sample of elderly?

Reviewer 3 Report

This study analyzed the associations among “usefulness”, “subjective norms”, attitude toward usage”, “behavior intention”, “safety”, “social image” and “the usage of walking aids” and an impact of “image of social image” on the “use of walking aids”. This study provides clue to understand psychological feature of middle-aged adults and elderly and could be useful for health promotion in aged society. However, there are problems to be published in this study.

Introduction

The “Introduction” section is too long. The “Introduction” section must include the background and objective of the study, but it is difficult to understand what is the objective of this study. Explanations for “Design and development of walking aid”, “social image”, “Theory of Reasoned Action”, “safety”, “usefulness” and “Mediation effect” should be shortened.

Tables

P12-13 Table 3 In the text, “All composite reliability were between 0.875 and 0.964. But in the Table, a term “Constructed Reliability” is used. Which is correct?

P13 Table 4 It is hard to understand. Are numbers in bold square root of average variance?

P14 Table 6 In the column of “DV7”, the third to five line is blank. Does “BI” applied to them?

Abbreviations should be united. For example, in Table 4 “ATU”, but in Figure 4, “ATTU”.

Table3, 6 It is not exact that p-value is 0.000. It should be p<0.001.

Discussion

P13 line 496-499 Further discussion is necessary. An impact of social image toward usage of walking aids is the key part of this study.

4.2 practical Implication This section is too long. This part is authors’ proposal to the government based on the evidence obtained from this study, and should be kept to the minimum.

English

The authors should take professional English correcting service by native speaker. There are many expressions that are difficult to understand. For example, “because of others’ eyes (p1 line 13)”. Does this expression mean that many elderly worry about that others may think of them?

P10 line 385-387, P16 line 486-487, line 494 have great inconsistent opinions

I cannot understand the meaning of these sentences.

The results of previous studies → the results obtained from previous studies

Line 504 The word “conform” means “fit”?

Round 2

Reviewer 1 Report

The authors have improved the paper during the review process, implementing the main indications provided.

From my side, I recommend the publication.

Best Regards

Author Response

Thank you for giving us this opportunity to submit our manuscript titled “Social image impacting attitudes of middle-aged and elderly people toward the usage of walking aids: An empirical investigation in Taiwan”. We appreciate the time and effort that you and the reviewers have dedicated to providing your valuable feedback on our manuscript. Thank you very much for all the help.

Reviewer 3 Report

Most of problems are corrected and improved. However, there are some problems that are not improved. The authors need to manage the problems sincerely.

I cannot find reason why full explanation of “Design and development of walking aid”, “social image”, “Theory of Reasoned Action”, “safety”, “usefulness” and “Mediation effect” are required in an original article and have not seen original article explaining definition or something extensively.

I know “P=0.000” is presented in software. But it is just a notation by software. Academically, “P=0.000” is impossible. Researcher must understand the meaning of figures.

I have suggested to receive professional English correcting service by native speaker. Researchers in the world will read this article when it is published.

Author Response

Dear Reviewer

Thank you for giving us this opportunity to submit a revised draft of our manuscript titled “Social image impacting attitudes of middle-aged and elderly people toward the usage of walking aids: An empirical investigation in Taiwan”. We appreciate the time and effort that you have dedicated to providing your valuable feedback on our manuscript. We have highlighted the changes within the manuscript. Here is a point-by-point response to the reviewer’ comments and concerns.

Comment 1:

Most of problems are corrected and improved. However, there are some problems that are not improved. The authors need to manage the problems sincerely.

Response: Thank you for this reminding, we will do our best to reply all comments sincerely.

Comment 2:

I cannot find reason why full explanation of “Design and development of walking aid”, “social image”, “Theory of Reasoned Action”, “safety”, “usefulness” and “Mediation effect” are required in an original article and have not seen original article explaining definition or something extensively.

Response: Thank you for the suggestion. We have done the modifications and shortened the sections of the “Design and development of walking aid” (which was descripted from line 139 to line158), “social image” (which was descripted from line 159 to line 191), “Theory of Reasoned Action” (which was descripted from line 197 to line 232), “safety” (which was descripted from line 233 to line 254), “usefulness” (which was descripted from line 255 to line 273) and “Mediation effect” (which was descripted from line 274 to line 288). Additionally, Figure 3 detailed the research model of this study was also considered grounded because it is a reflection of the research questions, framework of inquiry, including variables. The section of 3.2 Measurement instrument was shown more descriptions about the users' attitude toward using walking aids from line 314 to line 337. Hopefully, the above descriptions we provide will help us to reply the reviewer's comments attached with this revision.

Comment 3:

I know “P=0.000” is presented in software. But it is just a notation by software. Academically, “P=0.000” is impossible. Researcher must understand the meaning of figures.

Response: Thank you for reminding us again. We corrected P value from P=0.000 to P<0.001 at table 3, 7.

Comment 4:

I have suggested to receive professional English correcting service by native speaker. Researchers in the world will read this article when it is published.

Response: Thank you for your suggestion. We corrected the sentences in line 127-128, 135-136,180-183, 282-283.

The reviewer’ comments were highly insightful and enabled us to greatly improve the quality of our manuscript. All the lines indicated above are in the revised manuscript. We thank the reviewer for the kind advices. We hope that the revisions in the manuscript and our accompanying responses will be sufficient to make our manuscript suitable for publication in the Journal of Healthcare.
